# Systematic Comparison of FBS and Medium Variation Effect on Key Cellular Processes Using Morphological Profiling

**DOI:** 10.3390/cells14050336

**Published:** 2025-02-25

**Authors:** Timofey Lebedev, Alesya Mikheeva, Valentina Gasca, Pavel Spirin, Vladimir Prassolov

**Affiliations:** Engelhardt Institute of Molecular Biology, Russian Academy of Sciences, 119991 Moscow, Russia; alesyamikheeva@gmail.com (A.M.); valentinagasca@gmail.com (V.G.); spirin@eimb.ru (P.S.); prassolov45@mail.ru (V.P.)

**Keywords:** cell culture, growth conditions, automated microscopy, drug sensitivity, cell morphology

## Abstract

Although every cell biologist knows the importance of selecting the right growth conditions and it is well known that the composition of growth medium may vary depending on a product brand or lot affecting many cellular processes, still those effects are poorly systematized. We addressed this issue by comparing the effect of 12 fetal bovine sera (FBS) and eight growth media from different brands on the morphological and functional parameters of five cell types: lung adenocarcinoma, neuroblastoma, glioblastoma, embryonic kidney, and colorectal cancer cells. Using high-throughput imaging, we compared cell proliferation; performed morphological profiling based on the imaging of 561,519 cells; measured extracellular regulated kinases (ERK1/2) activity, mitochondria potential, and lysosome accumulation; and compared cell sensitivity to drugs, response to EGF stimulation, and ability to differentiate. We found that changes in cell proliferation and morphology were independent, and morphological changes were associated with differences in mitochondria potential or the cell’s ability to differentiate. Surprisingly, the most drastic differences were detected in serum-free conditions, where medium choice affected cell survival and response to EGF. Overall, our data may be used to improve the reproducibility of experiments involving cell cultures, and the effects of 28 growth conditions on proliferation and 44 morphological parameters can be explored through a Shinyapp.

## 1. Introduction

Stable human cell cultures are an essential tool for fundamental and translational biology research, and they often provide the basis for majority of cell biology studies [1,2]. The development of optimized growth media compositions and serum production standards facilitated the use of cell cultures in research and allowed to standardize this procedure [2]. However, the ongoing problem is a low reproducibility of scientific discoveries, including those obtained using stably passaged cells [3]. Low reproducibility may arise from various reasons such as differences in experimental protocols, data processing and interpretation [3], incorrect application of statistical methods [4], cell lines contaminations with other cell types [5] or mycoplasma [6], and the misidentification of cell lines [7] but also due to varying cell growth conditions [8]. Although most basal growth media are synthetic with well-established formulations, there are still factors affecting medium performance depending on the manufacturer or particular lot. These factors include different sources of medium components, water quality, concentrations of vitamins, glucose, and the choice of essential amino acids, and various additives such as HEPES to improve the buffer capacity of the medium. Most studies also use animal-derived serums, such as fetal bovine serum (FBS), since synthetic serums or protein-free analogues currently do not match the versatility and costs of animal-derived serum. However, animal-derived serums are infamous for having an unknown and varying composition, batch-to-batch differences, and being affected by locations and animal conditions at the site of product collection [9,10,11,12,13].

The most prominent and well-known effects the serum and basal medium selection have are on cell proliferation and morphology [14,15,16]. Hormones present in serum have a direct effect on the growth of hormone-sensitive cells, such as breast cancer cell lines [17]. Other cellular processes are also directly affected by serum choice: for example, FBS from different brands can have drastic effects on cytokine secretion, such as IL-8 [8,18], the ability of carbonyl cyanide m-chlorophenyl hydrazone (CCCP) to depolarize mitochondria depends on the serum levels in culture media [19], serum concentration affects mesenchymal stem cell immunosuppressive properties [20], and manganese level variations in serum affect glycosylation defects [21]. Serum concentration and media choice can mildly affect cell viability readouts, drug efficacy [22,23], and cell differentiation [24,25]. The high glucose and pyruvate concentrations used for most basal media may repress mitochondrial respiration and impact replication levels of the studied viruses [26,27]. Serum composition affects the lipopolysaccharide (LPS)- and zymosan-induced procoagulant activity of mononuclear cells [28], while FBS has bactericidal activity against *Helicobacter pylori* [29], probably due to proteins of complement system, and it can enhance the antibacterial activity of cationic and amphiphilic polymers [30]. The medium composition can even impact CRISPR screens for essential genes, and the use of more physiological medium reveals new cancer dependencies, especially among genes involved in glycolysis (HK2, GPI, and PFKP), metabolites transporters (SLC25A1, SLC25A11, and SLC7A1), mTOR signaling, and even translation initiation factor EIF1 and antiapoptotic factor BCL2L1 [31].

The selection of appropriate serum/media is also a major factor contributing to experimental reproducibility. Also, the laboratory aimed at replicating essential results from those studies might not have access to the exact same reagents. High-throughput screening studies for cell lines, such as DepMap [32], Genomics of Drug Sensitivity in Cancer [33], Cancer Target Discovery and Development (CTD^2^) Network [34], CellPainting [35], and Dye Drop [36] are becoming an essential tool for basic and fundamental research. They often utilize large numbers of cell lines and collect data over many years. Due to the studies’ scale, it is often impossible to use the same growth media and serum lots or even brands within such studies. Despite increasing evidence that serum/media selection affects cell morphology, the exact data on these effects are very limited [14]. In our study, we aimed to provide a comprehensive comparison how basal growth media and serums from different brands affected various cellular processes. We chose eight different media (each DMEM and RPMI-1640) and twelve FBS brands, as the most used type of serum, from different manufacturers across the world and selected cell lines which belong to different tissues and represent some of the most common studied types. Lung cancer H1299 cells and colorectal HCT-116 cells belong to two of the most common solid cancers, glioblastoma LN-18 and neuroblastoma SH-SY5Y belong to most common cancers of neuroendocrine origin, and embryonic kidney cells HEK-293T is one of the most commonly used cell lines for gene function studies, protocol optimizations, virus production, protein secretion and many others. SH-SY5Y are also widely used in neurobiology studies, as they may be differentiated into neural-like cells with adrenergic properties [37]. The selected cell lines also grow in similar conditions, such as DMEM or RPMI-1640 with 10% FBS without the need for additional supplements, so we could provide a comparison between those cell lines. We used high-throughput cell imaging and image processing software to measure cell proliferation, changes in cell morphology, drug response, mitochondria activity, lysosomal accumulation, ERK1/2 and growth factor signaling, and cell differentiation.

## 2. Materials and Methods

### 2.1. Cell Cultures

Human lung adenocarcinoma H1299 (ATCC #CRL-5803), glioblastoma LN-18 (#CRL-2610), colorectal HCT-116 (ATCC #CCL-247), and embryonic kidney cells HEK-293T (ATCC #CRL-3216) cells were initially cultured in DMEM (#61965026, Gibco, Waltham, MA, USA) with 10% FBS (#10270106, Gibco, USA). Neuroblastoma SH-SY5Y (ATCC #CRL-2266) cells were cultured in RPMI-1640 with 10% FBS (#10270106, Gibco, USA). H1299, LN-18, HCT-116, and SH-SY5Y cells had a stable expression of H2B-mRuby nuclear marker (Addgene # 90236) and ERK-KTR-mClover reporter (Addgene # 59150) [38]. Reporter cell lines were generated previously [39,40]. HEK-293T with H2B-mRuby and ERK-KTR-mClover reporter expression were generated by lentiviral transduction as described previously [39]. Additional DMEM, RPMI-1640 and FBS were purchased from Sigma Aldrich (St. Louis, MO, USA), Corning (New York, NY, USA), Capricorn Scientific (Ebsdorfergrund, Germany), Cytiva (Marlborough, MA, USA), Servicebio (Wuhan, China), Globe Kang (Qinhuangdao City, China), HiMedia (Maharashtra, India), Biosera (Cholet, France), BioinnLabs (Rostov-na-Donu, Russia), PanEco Ltd. (Moscow, Russia), and Dia-M (Moscow, Russia). Detailed information of growth conditions and reference numbers for all products are provided in Table 1 and Appendix A. Complete growth media was also supplemented with 1 mM sodium pyruvate (#11360070, Gibco, Waltham, MA, USA), 2 mM L-glutamine (#25030081, Gibco, Waltham, MA, USA), and 100 U/mL penicillin–streptomycin (#15070063, Gibco, Waltham, MA, USA), if those supplements were not already included in media formulation by manufacturer. Cells were passaged 2–3 times after thawing and before the start of experiments to ensure stable proliferation. For the experiments, cells were maintained for not more than for 3 additional passages. All initial cell lines were a gift from Prof. Dr. Carol Stocking from Heinrich-Pette Institute for Experimental Virology, Hamburg, Germany. None of the used cell lines is listed in the list of commonly misidentified cell lines maintained by the International Cell Line Authentication Committee. Mycoplasma was tested after cell thawing and for every two weeks of cultivation using DAPI and Hoechst-33342 staining to ensure mycoplasma-free cultures.

### 2.2. Cell Imaging

Cells were plated in 96-well plates (1 × 10^3^ H1299, 1.5 × 10^3^ LN-18, 2.5 × 10^3^ HEK-293T, 2 × 10^3^ HCT-116, and 2.5 × 10^3^ SH-SY5Y) in their initial complete growth media. After 24 h, the initial media was removed and media/sera from different brands were added. Cells were imaged 24 h and 72 h after growth media change in bright field and 460–500 nm excitation/512–542 nm emission and 541–551/565–605 nm channels for ERK-KTR-mClover and H2B-mRuby fluorescence using 10× objective. Images were taken using a Leica DMI8 (Wetzlar, Germany) automated microscope. The endpoint of 72 h for measuring cell proliferation was selected, because near 72 h, most cell lines reach 70–80% confluence, which is usually an optimal density for replating and such density does not affect cell segmentation. Experiments were performed in at least three independent repeats; for each repeat, four fields per well were imaged using automated field selection and autofocus.

### 2.3. Morphological Profiling and Cell Proliferation Analysis

Cell segmentation was performed using bright-field images with a cyto3 model from Cellpose v3 [41]. Nucleus segmentation was performed using H2B-mRuby images in CellProfiler v4.2.5 [42] with prior illumination correction. Then, cell and nucleus objects were matched so only nuclei with 90% overlap with the cell object were selected and only cells with a nucleus object within them were used further for analysis. Images with quality issues, such as bad focus, optical obstacles, serum debris or unexplained large variations in cell density were removed if the number of identified objects or distribution of morphological features substantially differed from all other images for the experimental condition, as described in [40]. The low quality of removed images was confirmed manually by several researchers. For analysis of SH-SY5Y cell differentiation, nuclei identified from H2B-mRuby images were used as seed objects and ERK-KTR-mClover images were used to measure cell branching and the length of the morphological skeleton. Cells with cytoskeleton length > 100 pixels were considered as differentiated cells (mean length for undifferentiated cells was 19 pixels).

For each growth condition, we obtained 123 morphological parameters (Appendix A) measured for several hundred or thousands of cells. To compare the changes in the measured morphological features, we first normalized each feature distribution across all measured cells using a z-score. The distribution of morphological parameters in cell population is not typically Gaussian, so we calculated Wasserstein distance, which is also called earth mover’s distance (EMD). This metric depends not only on the differences between distribution medians but also depends on differences in distribution shapes. For each pair of conditions, we compared distributions for each of 123 parameters, which gave us a vector consisting of 123 Wasserstein distances. Then, we calculated Euclidean distances based on those vectors to measure the dissimilarity between each pair of growth conditions. The dimension reduction was applied so we could visualize those precomputed dissimilarities in a 2D space.

Filtered cell objects that had a corresponding nucleus were used for cell counting. Cells were counted for each image; then, the mean number of cells for each repeat was calculated based on two to four images that passed an image quality check. To account for variations across wells plated on different days, the cell numbers for each condition were normalized to the number of cells in a control well (where cells grew without medium change) at the same date. Growth rate was calculated as the ratio of cells at 72 h to the number of cells at 24 h in the same well.

### 2.4. ERK1/2 Activity Measurement

ERK1/2 activity was measured by ERK-KTR-mClover reporter (Addgene # 59150) [38]. For segmented and filtered cells, we calculated the reporter intensity in previously identified nuclei and the part of cytoplasm defined as a 15-pixel ring around the nucleus. The ERK1/2 activity was calculated as a ratio between the median reporter fluorescence intensity in a part of the cytoplasm to the intensity in the nucleus (C/N ratio) in CellProfiler. Mean ratios for each image were normalized, so 0 represents the minimal observed ERK1/2 activity (fully inhibited ERK1/2 with selective inhibitor) and 1 represents the maximum observed activity (when cells are stimulated by a growth factor). Normalization was performed using reporter values from a previous study, where reporter functionality was described for cell lines used in this study. For measuring ERK1/2 activation with EGF, H1299 cells were seeded in media with 10% FBS, and after 24 h, the media was changed on serum-free media, and after an additional 24 h, we added either 100 ng/mL recombinant human EGF (ab259398, Abcam, Cambridge, UK) or 0.001% BSA for control. The ERK1/2 activity was measured with an ERK-KTR-mClover reporter for 30 min after EGF addition.

### 2.5. Mitochondria and Lysosome Staining

H1299 cells without fluorescent reporter expression were stained with 100 nM lysosomal stain LumiTracker LysoGreen (Lumiprobe, Moscow, Russia) and potential-dependent mitochondrial stain 100 nM TMRE (Lumiprobe, Russia). For nucleus and cell segmentation cells, we stained cells with 1 µg/mL Hoechst-33342 (Sigma Aldrich, USA) to image nuclei and 1 µM Tubulin Tracker Deep Red (Invitrogen, Waltham, MA, USA) to image the cytoskeleton. Hoechst-33342, Tubulin Tracker and TMRE were added to the growth medium for 30 min, and then LysoGreen was added for 5 min. After incubation with stains, cells were washed with serum-free DMEM without phenol red (BioinnLabs, Russia) and then imaged on a Leica DMI-8 fluorescent microscope. All fluorescent images were corrected for uneven illumination in CellProfiler prior to any segmentations. Nuclei were segmented using a cyto3 model from Cellpose v3 for Hoechst-33342 images, and then cells were segmented with propagation algorithms in CellProfiler v4.2.5 using merged Tubulin and TMRE images and nucleus objects as seeds. Integrated and mean intensities of TMRE and LysoGreen for each cell were calculated and then normalized to staining intensities in control cells to measure the mitochondria activity and lysosomal accumulation.

### 2.6. Drug Treatment and Cell Viability Measurements

Cells were seeded in the initial growth media; then, 24 h later, the media was changed, and cells were treated with 1–20 µM sorafenib (Shanghai Macklin Biochemical, Shanghai, China), 1–20 µM etoposide (Selleck Chemicals, Houston, TX, USA), 10–1000 nM paclitaxel (Shanghai Macklin Biochemical, China), and 0.1–10 µM palbociclib (Selleck Chemicals, Houston, TX, USA). All drugs were dissolved in DMSO, and the final DMSO concentration for each treatment was 0.1%, including control cells treated only with DMSO. Cell viability was analyzed after 72 h incubation with drugs using an AbiCell Resazurin Cytotoxicity Assay Kit (Abisense, Moscow, Russia) on a Multiscan FC (Thermo Scientific, Waltham, MA, USA) plate reader. The shift in absorption at 620 nm to 570 absorption was measured to calculate the resazurin conversion. Measurements were normalized, so the absorption difference for DMSO-treated control cells was used as 100% and the difference in wells without cells was used as a baseline: 0%. For cell differentiation, SH-SY5Y cells were seeded in initial media with 10% FBS at 1000 cells per well in a 96-well plate. Then, 24 h later, the cells were washed with PBS, and media containing 5% of tested FBS and 10 µM retinoic acid (Sigma-Aldrich, St. Louis, MO, USA) was added. After 96 h incubation with retinoic acid, the media was renewed, and fresh 10 µM retinoic acid was added. After an additional 96 h (total of 168 h) ERK-KTR-mClover, H2B-mRuby and bright-field images were taken using a Leica DMI-8 fluorescent microscope.

### 2.7. Data and Statistics Analysis

Data processing, statistical analysis and data visualization were performed using Python 3.11.5 and R 4.2.3. Heatmaps were generated using a ComplexHeatmap package 2.22.0 [43], and images were processed using Cellpose v3 [41] and CellProfiler v4.2.5 [42]. A comparison of morphological features and image quality control was performed using the Wasserstein metric and z-scores. The statistical significance of the observed differences in features between conditions was analyzed using the mean data for each image and a non-parametric Kruskal–Wallis test. The CellProfiler and Cellpose pipelines and the Python code for data processing are available at https://github.com/CancerCellBiology/FBS-and-Medium-morphology-screen (accessed on 30 January 2025).

## 3. Results

### 3.1. Cell Proliferation

To test the variance in cell proliferation and key cellular processes introduced by different growth media and serums, we selected 8 common media M1–M8 (both DMEM and RPMI) and 12 fetal bovine serums S1–S12 (FBS) from different manufacturers (Table 1, Figure 1). Initially, all cell lines were cultured in DMEM or RPMI (for SH-SY5Y cells) labeled M1 with 10% of FBS labeled S1 (referred to as S1/M1). First, we plated cells in a 96-well plate in medium M1 and 10% FBS S1, and 24 h later, the medium was discarded and different combinations of medium and serum were added. Then, we varied either the serum or medium and measured the cell growth under three types of conditions: different serums S1–S12 were mixed with the same medium M1, different media M1–M8 were mixed with the same serum M1, and additionally, we tested media M1–M8 without any serum. To measure the cell proliferation, ERK1/2 activity and cell morphology changes, we used cells with fluorescent nuclear marker H2B-mRuby and ERK1/2 reporter ERK-KTR-mClover. Cells were imaged 24 h and 72 h after media change in bright-field and fluorescent channels. Bright-field and H2B-mRuby images were used to count the number of cells under different growth conditions, and these images were used to quantify the changes in cell and nuclei shapes using Cellpose v3 [41] and CellProfiler v4.2.5 [42]. To account for the technical variations in cell seeding densities and rates of proliferation on different days, each time, we also performed experiments under control conditions without changing the growth media. Since experiment repeats and some conditions were tested on different days, we normalized cell proliferation to a control on the same day. The cell proliferation in control wells during different experiment days was stable and showed little variance, ensuring the high reproducibility of the measurements (Appendix A). All cell proliferation data are available at https://lebedevtdeimb.shinyapps.io/FBSMediumBrowser/.

Most serums and media with 10% FBS were able to provide cell proliferation rates comparable with controls. We noticed some variation in the number of cells at 24 h after medium change, which was either due to cell adaptation to new growth conditions or due to cell loss during medium change; however, the cell proliferation rates were the same for the most conditions. To address this issue, we compared conditions by proliferation rates, which were calculated as a fold increase in the cell numbers at 72 h to 24 h. Serums S1, S4, S5, S6, and S10 sustained high proliferation rates for all cell lines, marking them as having a more universal composition. In comparison, S8 and S11 had a moderate effect on HEK-293T cells, while S3, S7, S9, and S12 had a weaker effect on the proliferation of HCT-116 or SH-SY5Y cells, but they stimulated the growth of HEK293T, H1299 and LN-18 to the same levels as other serums (Figure 2a). S2 had the highest effect on HCT-116, but it had a moderate effect on the growth of other cell lines and had the weakest effect on SH-SY5Y, suggesting that this serum has the most diverging composition of mitogenic factors.

Although the serum provides the most pro-mitogenic factors, the change in growth medium also caused changes in cell proliferation rates (Figure 2b). As with the serum, H1299, LN-18, and HEK293T were the least affected by different media, with M1, M3, M4, and M8 provided the fastest proliferation. M6 and M7 failed to promote HCT-116 cell growth, and M2 failed to promote LN-18 growth. The variations caused by growth media were much more apparent in serum-free conditions (Figure 2c). All cell lines except SH-SY5Y were able to proliferate for 72 h without serum although at lower rates compared to 10% FBS. HCT-166 and LN-18 proliferation was most dependent in the medium used, as M5, M6, and M7 were not able to sustain their proliferation and had a weaker effect on H1299 and HEK-293T. M4 and M8 overall sustained higher proliferation rates for all cell lines, suggesting that these media might have some pro-mitogenic additives. Only three media (M3, M4, and M8) sustained SH-SY5Y survival for 72 h, while with other media, cell numbers declined. Additional plots showing data variance are provided in Appendix A. These show the high cell-specific dependency of proliferation on medium composition in the absence of serum.

### 3.2. Morphological Changes

Although the main criteria for choosing growth medium and serum is the ability to sustain cell proliferation, in vitro growth condition can affect other cellular process not directly related to proliferation. To address other changes in cell behavior, we measured cell and nucleus morphology by processing bright-field images for cell shapes and granularity, and we measured H2B-mRuby for nuclei parameters (Figure 3a, Appendix A). Overall, we analyzed 3553 images for all cell lines and collected data for 561,519 individual cells. In total, we measured 125 features for each cell and their nucleus and used Wasserstein metrics to compare features distributions between each condition and respective control. We focused on morphology changes after 72 h cultivation, as they were more pronounced; for example, several growth conditions changed cell and nucleus sizes as well as cell granularity (Figure 3a,b). In total, 44 features had statistically significant differences for at least one growth condition, as analyzed by the Kruskal–Wallis test. Differences in morphology were observed also for conditions that sustained similar cell proliferation rates and the same cell density. For example, cells cultured with serums S9 and S12 had a larger cell radius than cells cultured with S6, which caused a higher granularity. Similarly, conditions that resulted in different cell densities can have a similar morphology. Despite that, cells cultured in medium M4 had 1.8 higher density than cells in medium M6 (Figure 2b); they displayed very similar morphology.

To obtain an overview of similarities and difference in cell morphology induced by different growth conditions, we calculated the Euclidean distances between each pair of conditions in the n-dimensional space of Wasserstein distances and then used dimension reduction to visualize the results in 2D space. Generally, as expected for all cell lines, the largest difference was observed between complete and serum-free conditions, which was probably due to the differences in cell proliferation and cell death induced by serum starvation: for example, as shown for H1299 cells cultured in medium M6 without serum (Figure 3b,c). The morphology difference between cells maintained in control conditions and seeded on different days was considerably lower than some of the differences induced by medium or serum change. For most cell lines, a change in medium or serum had a similar amplitude of effect on the morphology, except HCT-116 cells, for which the medium change had a more pronounced effect on the morphology than the serum change. The distribution for 44 morphological parameters for each condition can be viewed at https://lebedevtdeimb.shinyapps.io/FBSMediumBrowser/.

### 3.3. ERK1/2 Activity and Response to EGF Stimulation

The MAPK–ERK signaling pathway is a major regulator of cell proliferation and survival in response to different mitogens. Serums might contain varying amounts of growth factors, which in turn might affect the MAPK signaling activity. The main component of MAPK signaling is ERK1/2 kinases, whose activation controls transcriptional programs induced by growth factors. The cell lines used in this study express an ERK1/2 activity reporter (ERK-KTR-mClover), which allows the measurement of ERK1/2 activity using fluorescent microscopy by comparing its fluorescence intensity in the cytoplasm and nucleus. We used this technique to compare how growth conditions affected ERK1/2 activity 24 h after growth media change. Under most conditions, the ERK1/2 activity changes were mild, and we expect a few cases did not exceed 10% of that total kinase activity range (Figure 4a). Serum removal affected ERK1/2 the most, as serum-free growth conditions clustered together. Initially, we expected that serum starvation would lead to lower ERK1/2 activity; however, we detected cell type-dependent changes in ERK1/2 activity. For SH-SY5Y cells, the ERK1/2 activity for serum-free conditions was lower than for full growth media; however, for HEK-293T and to a lesser extent for H1299, serum-free conditions upregulated ERK1/2 activity. LN-18 and HCT-116 did not display any noticeable differences in ERK1/2 activity. Notably, a group of media M2, M5, M6, and M7 demonstrated a distinct pattern of ERK1/2 as it increased ERK1/2 activity in HEK-293T and SH-SY5Y cells and decreased in H1299 cells (Figure 4a). Those same media sustained lower proliferation rates but a higher overall density of cells (Figure 2b), meaning that higher ERK1/2 activity may contribute to faster cell adaptation after seeding and reaching a monolayer. Although M2, M5, and M6 also sustained activated ERK1/2 without serum, only M2 was able to stimulate cell proliferation (Figure 2c). On the contrary, M4 and M8 were the only media that increased ERK1/2 activity in HCT-116 cells without serum, and both media maintained the highest cell proliferation among serum-free conditions. Serums S7 and S3 had the weakest proliferation stimulation and failed to maintain ERK1/2 activity at the same level as other serums for HCT-116 cells.

Another important aspect of ERK1/2 signaling is the ability of growth factors to activate this signaling pathway. Usually, to avoid interference from other growth factors, which may be contained in serum, growth factor signaling is studied in serum-free conditions. To test how cultivation in serum-free conditions affected growth factor signaling measurements, we measured the EGF effects ERK activation. H1299 cells were maintained in the respective medium without FBS for 24 h, and then cells were treated with 100 ng/mL EGF for 30 min (Figure 4b). For all media, we detected ERK1/2 activation caused by EGF addition, and it was most pronounced for M2 and M1, and it was least pronounced for M5 and M6 (Figure 4c). Short-term serum starvation in M5 and M6 media also resulted in lower ERK1/2 activity without EGF, and these media could not effectively stimulate H1299 proliferation without serum.

### 3.4. Drug Sensitivity

Next, we tested how growth conditions affected cancer cell sensitivity to commonly used anticancer drugs. We selected four drugs with different mechanisms of action: MEK1/2 inhibitor selumentinib, CDK4/6 inhibitor palbociclib, DNA synthesis inhibitor etoposide, and microtubule polymerization inhibitor paclitaxel. For the test, we selected six serums S1, S3, S4, S7, S8, and S9 that had different effects on H1299 cell morphology and differently stimulated cell proliferation. Cell viability was measured using a widespread resazurin test. For all drugs expect palbociclib, we did not detect any significant differences between drug responses, and even for palbociclib, those differences were observed mainly for lower concentrations (Figure 5a).

### 3.5. Mitochondrial Potential and Lysosomes Accumulation

Mitochondria and lysosomes maintain important processes for cell homeostasis, proliferation, survival, response to drugs, and differentiation. These organelles are frequently studied in different aspects of cell biology, and they can be directly affected by changes in growth conditions. We studied the effects of serums on those organelles using a mitochondria potential-dependent fluorescent stain TMRE and lysosomal stain LysoGreen. After 72 h of culturing with different serums, H1299 cells were stained with TMRE and LysoGreen as well as with Hoechst-33342 and TubulinTracker DeepRed for cell segmentation, and then cell images were processed using CellPose and CellProfiler (Figure 5b). Cells under serum-free conditions had accumulated lysosomes and had slightly decreased mitochondria activity, which was probably due to cell adaptation to reduced growth conditions. Different serums had no effect on lysosomal accumulation, but S3, S7, and S9 increased mitochondrial activity compared to other serums. Notably, those serums resulted in less sensitivity to lower concentrations of palbociclib (Figure 5a), and S9 resulted in changed cell morphology with increased cell and nucleus sizes (Figure 3b).

### 3.6. SH-SY5Y Cell Differentiation

Another important aspect of cell biology is studies of cell differentiation. Cell differentiation protocols often require reduced serum concentrations to 2–5% FBS in order to sustain cell survival but reduce cell proliferation [44]. Using well-established protocols for SH-SY5Y differentiation with retinoic acid, we measured the effects of serum on cell differentiation. Cells were cultured in media with 5% of different FBS in the presence of 10 uM retinoic acid. Differentiation media was changed after 96 h, and cells were cultured for an additional 72 h and then imaged. We used previously established protocol to evaluate SH-SY5Y differentiation based on changes in cell morphology [45], and we defined differentiated cells as cells with a cytoskeleton length higher than 100 pixels (Appendix A). Neither serum fully prevented cell differentiation; however, the percentage of differentiated cells ranged from 19% for S3 to 34% for S9 (Figure 5c). The S3 FBS which yielded the lowest percentage of differentiated cells also caused the biggest differences in SH-SY5Y morphology under normal growth conditions (Figure 3c). Under all conditions, cells displayed similar morphology, which significantly differed from control cell cultured with 5% FBS but without retinoic acid.

## 4. Discussion

As expected, serum and media from different brands had cell-type dependent effects on cell proliferation, and the proliferation of some cell lines, such as HCT-116, was more dependent on serum/growth media. Despite the fact that media like DMEM or RPMI-1640 are synthetic, do not contain growth factors or animal-derived components, and should be much more standardized than animal-derived FBS, we found the same level of variations in cell proliferation and morphology caused by changes in the DMEM/RPMI-1640 brand and the FBS brand. Still, the growth media and FBS from almost all brands, except a few cases for HCT-116 and SH-SY5Y cells, were able to stimulate considerable proliferation, and they likely are able to sustain cell culture growth over time. Unexpectedly, we observed the most variance in cell proliferation and survival when cells were cultured under serum-free conditions, and some media performed differently in the presence or absence of serum. For example, media M5, M6, and M7 maintained a proliferation of cells in the presence of 10% FBS at levels comparable with other media; however, in serum-free conditions, they performed significantly worse than other media. On the other hand, media from brands M4 and M8 stimulated cell proliferation better than any other brand both with and without 10% FBS. This means that basal media might vary in the content of some factors which are essential for cell survival only under stress conditions, such as serum starvation. For example, the choice of medium affects which genes are essential for cell survival, as has been shown in multiple CRISPR-Cas9 screens [31,46]. Serum-free growth conditions are used for the culturing of some cells sensitive to serum components; to avoid the activation of immune cells; and for cytokine secretion, differentiation, studies of cell stress, growth factor signaling and many other applications [47,48,49,50]. We also observed differences in how EGF was able to stimulate ERK1/2 activity depending on the medium brand. These findings highlight the importance of testing growth media components not only for complete growth conditions but also for reduced conditions.

Using morphological profiling, we showed that changes in cell morphology occur mostly independently from changes in cell proliferation. Even two growth conditions which both provide high levels of cell proliferation may result in different cell morphology. Although most changes were not dramatic, they still should be considered, as they might represent changes in other fundamental cell processes. For example, we observed that changes in the cell and nucleus shapes and sizes coincided with some changes in mitochondria activity. This aligns with a recent study that showed that cells cultured in DMEM and human plasma-like medium had different numbers of genes, while knockout affected cell compartment morphology [46]. The observed elevation of mitochondria activity depending on FBS brand is consistent with several studies showing serum and media effects on the maintenance of mitochondria activity [19,27]. Such morphological changes are especially important for studies utilizing morphological profiling or cell imaging, for which a change in growth media/serum lot or brand might yield batch effects and false-positive results.

Surprisingly, we had not seen major variations in ERK1/2 activity except for a minor increase in ERK1/2 activity when cells were cultured in certain media. Only for three out of five cell lines (colorectal HCT-116, glioblastoma LN-18 and neuroblastoma SH-SY5Y), 24 h serum starvation caused an expected decrease in ERK1/2 activity, while for embryonic HEK-293T cells and lung adenocarcinoma H1299 cells, ERK/12 activity was slightly increased after serum removal. This effect most likely does not depend on mutations in the MAPK pathway, as HCT-116 cells have an activating mutation in KRAS [51], SH-SY5Y cells have a constitutively active mutant ALK receptor [52], and H1299 cells have an NRAS mutation [53]. All these mutations contribute to constant activation of the MAPK pathway, and LN-18 and HEK-293T do not have known mutations affecting ERK1/2. Our data also highlight the cell-type specific roles of ERK1/2 activity in cell proliferation. For example, in lung adenocarcinoma cells, H1299 ERK1/2 activation by growth medium or serum is not related to proliferation, while for HEK-293T cells, increased EK1/2 activity may translate into better cell adaptation and growth, or it can be caused potentially by cell stress. For other cells, like colorectal cancer HCT-116 cells, increased or decreased ERK1/2 activity directly translates into higher or lower proliferation. Since serum starvation is often used to lower background mitogenic signaling, it should be considered that serum starvation might have different outcomes depending on the particular cell line. Although ERK1/2 changes were mostly minor, it is possible that prolonged changes in ERK1/2 activity might lead to changes in cell behavior, such as senescence [54], which is associated with increases in the nucleus or cytoplasm volume [55].

The FBS composition from particular brands is usually very poorly defined, which is partially due to variations caused by animal stock breeding conditions, such as feeding, climate and season variations. FBS contains more than a thousand components: for example multiple growth factors, such as EGF, IGF-1, TGF-β, FGFs, VEGF, PDGF [56], hormones, microelements, immunoglobulins, proteins of complement systems and others [57]. FBS affects the cell secretome [58,59,60] and at the same time cell response to cytokines, such as IL-8 [8], indicating the complex effect of FBS composition on paracrine and autocrine signaling in cell culture. FBS also varies in concentration of up to 40 elements, such as iron, zinc, magnesium and others, which may contribute to differences in antigenic expression (CD31, CD54, CD106) [13]. Different ATP levels of biologically available ATP may cause differences in purinergic signaling [61]. FBS variation has effects on both the cell transcriptome [62] and metabolome, and it specifically affects tyrosine, propanoate, cysteine and methionine metabolism [8]. Media also have a profound effect on gene expression and can affect the expression of *CD44*, *CDH1*, *VIM*, and *CD24*, which are involved in epithelial-to-mesenchymal transition (EMT) [62]. EMT directly affects cell morphology, and such effects on the transcriptome may contribute to some of the morphological changes observed in our study. Still, current studies provide limited insight into which exact components of medium or FBS affect particular signaling pathways. For example, even though human plasma-like medium (HPLM) resulted in K562 cells being dependent on glycolysis, compared media had similar levels of glucose, so such dependencies may be caused by a combination of multiple factors [31]. Since FBS has a complex and highly variable composition, and many components have either redundant or complementing functions, to identify which process are affected by which factor would require complex multivariate analysis, such as a thorough characterization of FBS composition (growth factors, hormones, microelements, metabolites, immunoglobulins, and etc.), coupled with measuring cell response at the transcriptomic, proteomic and metabolomic levels.

Serum variation had a minor effect on drug sensitivity. For three out of four drugs, we observed no significant variations in cell survival and only detected changes for CDK4/6 inhibitor palbociclib when it was used at lower concentrations. Similar findings were reported previously where media type or serum concentration variations had a minor effect on drug sensitivity compared with other factors [22]. Notably, sera which made cells insensitive to lower doses of palbociclib also increased mitochondria activity. Previously, we and others showed that palbociclib increases mitochondria activity in cancer cells [40,63,64], and serum levels can affect the efficacy of drugs targeting mitochondria [19]. Drugs, such as palbociclib, lead to increased cell sizes, which require more energy production by mitochondria, and thus cell survival is dependent on mitochondria activation. We speculate that some FBS may facilitate mitochondria activation needed for cell survival, which can be caused by differences in the serum ATP levels [61]. Serum deprivation directly affects many mitochondrial pathways, such as the mitochondrial electron transport chain, the oxidation of branched chain fatty acids and the citric acid cycle [60], and FBS maintains the balance of oxidant gene expression, preventing ROS accumulation, and it inhibits neomycin-induced apoptosis [65]. Notably, DMEM or RPMI may also contribute to increased mitochondria activity compared to human plasma-like medium (HPLM) and neglect some of the drug’s effect on mitochondria [66]. Media composition may even determine cell dependence on antiapoptotic *BCL2L1* gene expression [31]. Thus, we speculate that growth medium/serum variation might have stronger effects on the sensitivity to drugs that alter mitochondria function.

We show that cell imaging and morphological profiling using only cytoplasm and nuclei shape and size parameters, obtained with bright-field imaging and one nuclear fluorescent marker, might significantly facilitate the choice of optimal growth conditions. Cells with fluorescent-marked cell structures, such as nuclear marker H2B-mRuby, can be used as a platform to rapidly test new growth conditions, and inexpensive dyes such as Hoechst-33342 might be used as a substitute to obtain similar results without the need to generate transgene cells. However, such approaches may be limited to adherent cell cultures, as suspension cells grown throughout the volume of the well, not only on the surface, can aggregate in certain parts of the well, float on different levels, and form large clumps. This makes analysis of suspension cell proliferation using automated microscopy less reliable, and their analysis would require methodology optimization to account for their growth patterns and validation of how precise cell proliferation in suspension can be measured by microscopy. Our collected data, available at https://lebedevtdeimb.shinyapps.io/FBSMediumBrowser/, can be used for searching a substitute for a particular brand if it becomes unavailable or to see which other brands can be used without changing cellular processes too much. Comparing morphological features provides additional information on the similarity of cellular processes, and the selection of growth conditions that provide a comparable level of cell proliferation and minimally alter basic cell morphology should improve reproducibility.

## 5. Conclusions

Our study highlights several important factors which should be considered when optimizing growth conditions or trying to reproduce published results. Cellular processes are not equally affected by the choice of media or sera; thus, cells cultured using different products might have equal proliferation rates, but that does not guarantee reproducibility in other measurements, such as mitochondria activity or differentiation induction. The choice of basal media, which is commonly overlooked by researches, is equally important as the choice of serum, especially for serum-free experiments. As a possible solution to those problems, we propose measuring changes in cell morphology. We show that even using a minimalistic setup, such as bright field imaging and nuclei staining, provides a valuable tool for improving reproducibility in cell culture experiments, as those measurements can capture changes in cell proliferation and various cellular processes.

## Figures and Tables

**Figure 1 cells-14-00336-f001:**
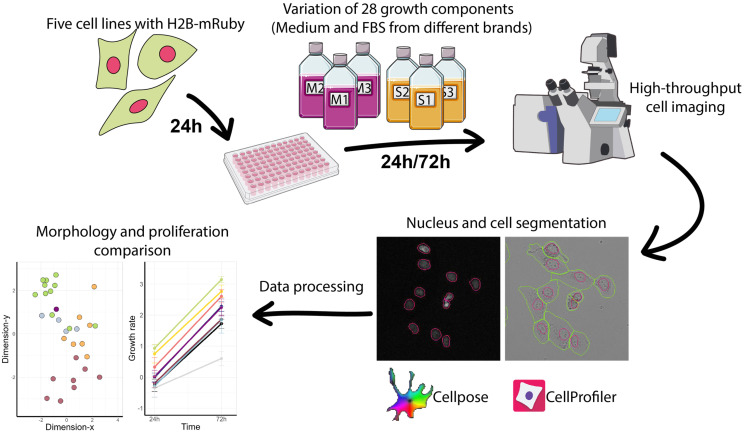
Experiment design. Several images were created using free available images: bottle-medium-pink icon by Servier https://smart.servier.com/ is licensed under CC-BY 3.0 Unported https://creativecommons.org/licenses/by/3.0/ (accessed on 7 October 2024); NIAID Visual & Medical Arts. Confocal Microscope. NIAID NIH BIOART Source. https://bioart.niaid.nih.gov/bioart/86 (accessed on 7 October 2024); 96_well_plate icon by Mar-cel Tisch is licensed under CC0 https://creativecommons.org/publicdomain/zero/1.0/ (accessed on 7 October 2024).

**Figure 2 cells-14-00336-f002:**
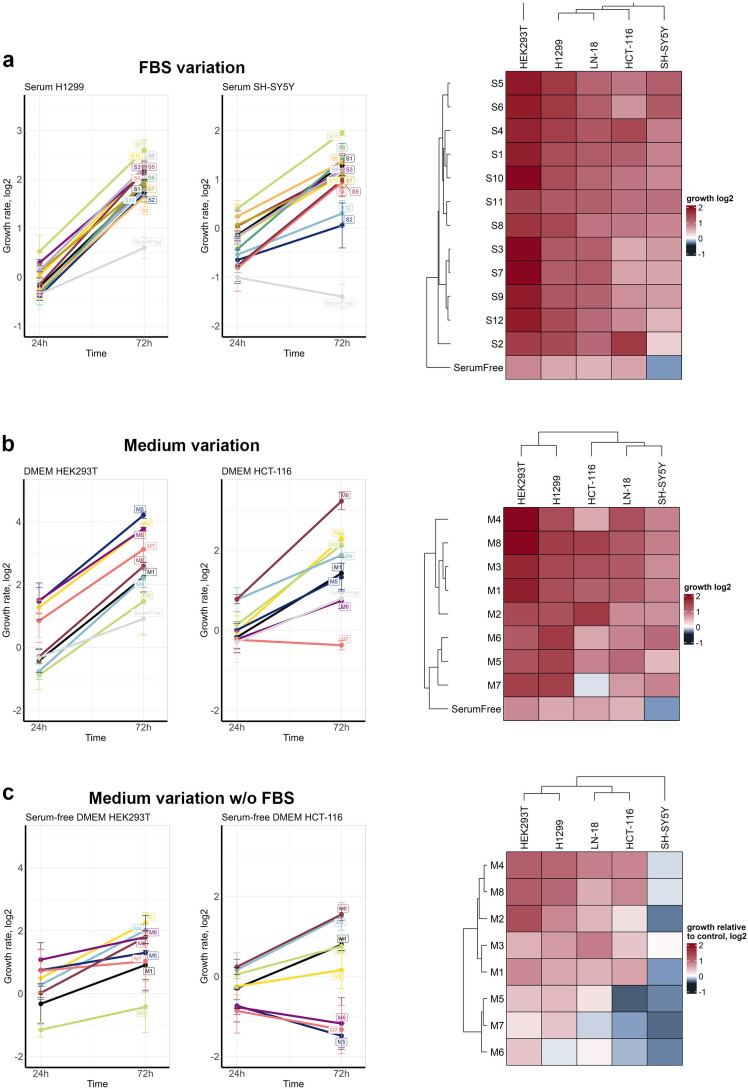
FBS and medium variation effect on cell proliferation. (**a**) FBS variation effects on cell proliferation, (**b**) basal media (DMEM or RPMI-1640) variation, and (**c**) media variation in serum-free conditions. Changes in growth rate are shown in log2 scale; the growth rate was calculated as the ratio of cell numbers at 72 h or 24 h relative to the control condition (without growth media change). Heatmaps show the clustering of growth conditions based on the proliferation of all cell lines. Growth rates for 72 h were used, and they were normalized across different cell lines, so for each cell line, a zero value is the absence of growth (same number of cells at 72 h as for 24 h), and value 1- is an average growth rate for each cell line. The Ward D2 algorithm with Euclidean metrics was used for heatmap clustering. SD values for three independent measurements are shown. Additional plots showing data variance are provided in Appendix A.

**Figure 3 cells-14-00336-f003:**
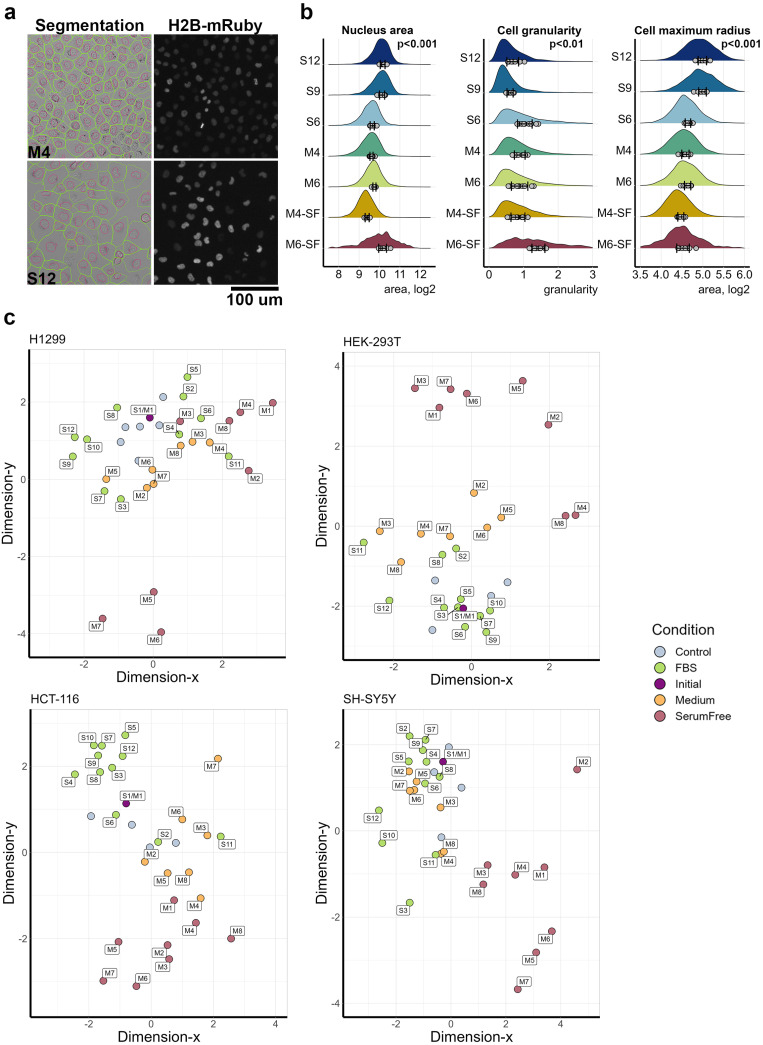
FBS and media selection affects cell morphology. (**a**) Example of cell border segmentation using bright-field images (green) and nucleus segmentation from H2B-mRuby images (red). (**b**) Changes in nucleus area, cell granularity, or cell maximum radius for selected conditions. The differences across conditions were analyzed using a non-parametric Kruskal–Wallis test. Mean values for each image are shown by gray dots, and the SD calculated for mean values is shown by bars. (**c**) Multidimensional reduction in distances between growth conditions calculated for each cell line using Wasserstein metrics for 124 morphological parameters. Control cells grown without media change; variation in control conditions at different days is shown by different data points.

**Figure 4 cells-14-00336-f004:**
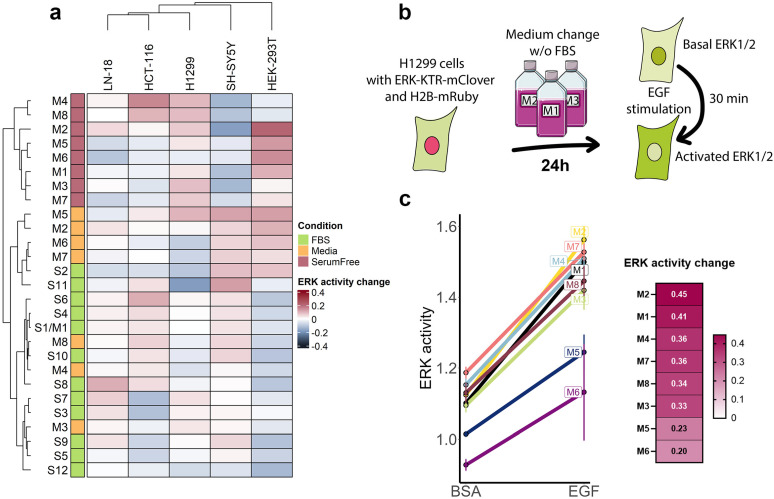
Changes in ERK1/2 signaling activity. (**a**) Heatmap of differences in ERK1/2 activity due to growth condition change. ERK1/2 activity change was calculated as a difference in normalized ERK-KTR-mClover reporter activity, so the difference equal to 1 represents the whole range of ERK1/2 activity. The Ward D2 algorithm with Euclidean metrics was used for heatmap clustering. (**b**) Experiment design for measuring cell response to EGF. (**c**) Changes in ERK1/2 activity between control wells (treated with 0.001% BSA) and 100 ng/mL EGF, which was shown as raw ERK-KTR-mClover activity values. Normalized ERK1/2 activity changes caused by EGF stimulation are shown on a heatmap. SD values for three independent measurements are shown.

**Figure 5 cells-14-00336-f005:**
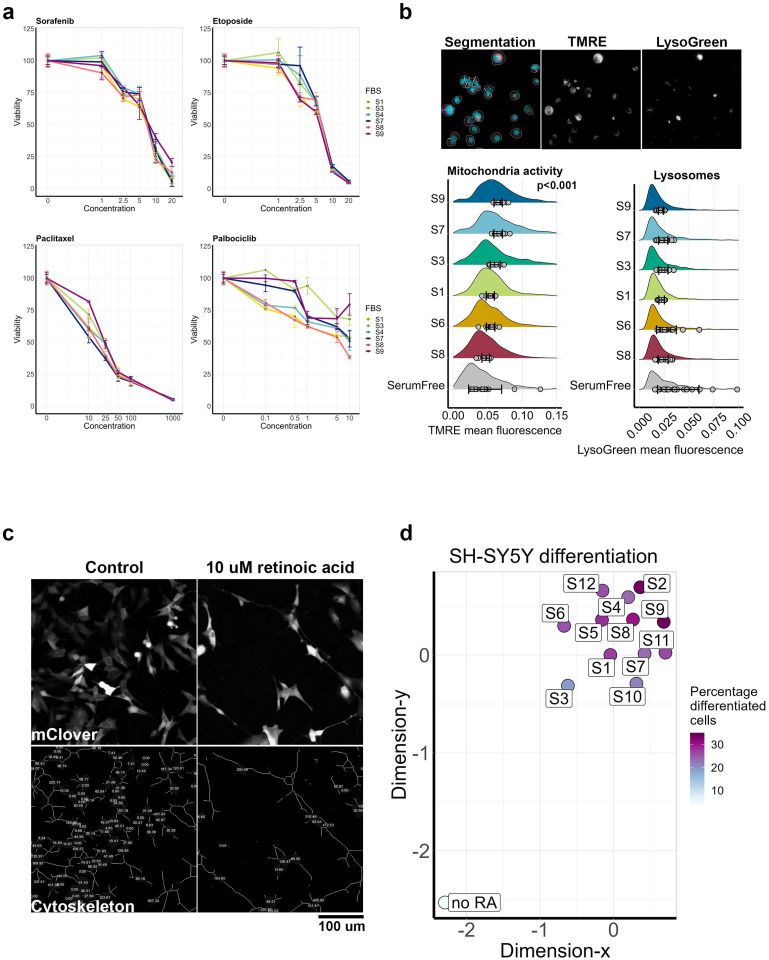
FBS and media selection affects drug sensitivity, mitochondria signaling and cell differentiation. (**a**) Cell viability of H1299 cells grown with six different FBS brands and treated with 1–20 µM sorafenib, 1–20 µM etoposide, 10–1000 nM paclitaxel, and 0.1–10 µM palbociclib for 72 h. Cell viability measurements were normalized to control, where cells were treated with DMSO, and the final DMSO concentration for each treatment was 0.1%. SD values are shown by bars and were calculated based on three repeats. (**b**) Cell staining with TMRE for mitochondria protentional and LysoGreen for accumulation of lysosomes. Distribution of mean fluorescence intensity per cell was compared using a non-parametric Kruskal–Wallis test. The mean values for each image are shown by gray dots, and the SD calculated for mean values is shown by bars. (**c**) SH-SY5Y cells imaged after 168 h treatment with 10 µM retinoic acid using ERK-KTR-mClover fluorescence and cytoskeleton images calculated in CellProfiler. Control cells were grown in 5% FBS without retinoic acid for 168 h. (**d**) Multidimensional reduction in distances between growth conditions calculated for SH-SY5Y cells differentiated with retinoic acid using Wasserstein metrics for 124 morphological parameters. Percentage of differentiated cells was indicated by color and was calculated as the percentage of the cell with a cytoskeleton length more than 100 pixels. Control cells (no RA) were grown in 5% FBS without retinoic acid for the same time.

**Table 1 cells-14-00336-t001:** List of FBS and media used in this study.

Code	Serum	Medium
S1/M1	10% FBS #10270106 (Gibco)	DMEM #61965026/RPMI-1640 #21870076 (Gibco)
M3	10% FBS #10270106 (Gibco)	DMEM #DMEM-HA/RPMI-1640 #RMPI-XA (Capricorn Scientific)
M4	10% FBS #10270106 (Gibco)	DMEM #SH30022.01/RPMI-1640 #SH30027.01 (Cytiva)
M5	10% FBS #10270106 (Gibco)	DMEM #AL066/RPMI-1640 #AL028 (HiMedia)
M6	10% FBS #10270106 (Gibco)	DMEM #C410E/RPMI-1640 #C310∏ (PanEco ltd)
M7	10% FBS #10270106 (Gibco)	DMEM G4511-500ML/RPMI-1640 #G4531-500ML (Servicebio)
M8	10% FBS #10270106 (Gibco)	DMEM #D0822/RPMI-1640 #R8758 (Sigma Aldrich)
S2	10% FBS #500SA (Dia-M)	DMEM #61965026/RPMI-1640 #21870076 (Gibco)
S3	10% FBS #FB-1200 (Biosera)	DMEM #61965026/RPMI-1640 #21870076 (Gibco)
S4	10% FBS #FBS-12B (Capricorn Scientific)	DMEM #61965026/RPMI-1640 #21870076 (Gibco)
S5	10% FBS heat-inactivated #FBS-HI-12A (Capricorn Scientific)	DMEM #61965026/RPMI-1640 #21870076 (Gibco)
S6	10% FBS #FBS-16A (Capricorn Scientific)	DMEM #61965026/RPMI-1640 #21870076 (Gibco)
S7	10% FBS #35-015-CV (Corning)	DMEM #61965026/RPMI-1640 #21870076 (Gibco)
S8	10% FBS #SH30088.03 (Cytiva)	DMEM #61965026/RPMI-1640 #21870076 (Gibco)
S9	10% FBS #RM10432 (HiMedia)	DMEM #61965026/RPMI-1640 #21870076 (Gibco)
S10	10% FBS heat-inactivated #RM9955 (HiMedia)	DMEM #61965026/RPMI-1640 #21870076 (Gibco)
S11	10% FBS #F800822 (Globe Kang)	DMEM #61965026/RPMI-1640 #21870076 (Gibco)
S12	10% Serum replacement #TCL280 (HiMedia)	DMEM #61965026/RPMI-1640 #21870076 (Gibco)

## Data Availability

All data for cell proliferation rates and most variable morphological features are available at https://lebedevtdeimb.shinyapps.io/FBSMediumBrowser/ (accessed on 31 January 2025). CellProfiler and Cellpose pipelines and python code for imaging data processing are available at https://github.com/CancerCellBiology/FBS-and-Medium-morphology-screen (accessed on 31 January 2025).

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
