# Peer review of "Systematic Comparison of FBS and Medium Variation Effect on Key Cellular Processes Using Morphological Profiling"

_cells, 2025, doi:10.3390/cells14050336_

Round 1

Reviewer 1 Report

Comments and Suggestions for Authors

In this paper, Lebedev et al. studies the effect of FBS from different conditions on affecting many morphological factors of several cell lines. I went through the paper and the following points need to be answered/investigated: 

  1. Since the number of passage is an importnat factor, the authors need to state how many times were the cell lines spiltted and recultured.
  2. What is the connection between these five cell types used in this study?
  3. Line 17: please provide the full name ofERK1/2.
  4. Line 55: what do the authors mean by CCCP?
  5. Line 57: please add a space before the word serum.
  6. Lines 60-62: this point is really intersting. Could the authors provide more details about which genes are affected?
  7. Since the authors mentioned that the medium affect the replication of viruses, could the authors provide some impact about the situation with bacteria?
  8. Why did the authors use only a single concentration of FBS in the study?
  9. All experiments were performed on cell lines, why there were no cell suspensions included?
  10. Was the expression of H2B-mRuby conditional or consecutive? how did the authors ensure that the cell variant phenotypes did not come because of the genetic expression of H2B-mRuby?
  11. Line 237: please add the word "to" before the word "provide".
  12. Lines 265-267: please indicate for how long time did the cell survive without FBS.
  13. I am still looking for a helpful conclusion of this study. What is the main message the authors want to deliver?

Author Response

We thank all reviewers for thorough review of our manuscript, suggesting interesting questions, correcting some mistakes and typos, and highlighting areas which were not sufficiently covered in the initial manuscript.  Here we provide point-to-point response to all questions. All changes in the manuscript are highlighted by color.

Comment 1: Since the number of passage is an importnat factor, the authors need to state how many times were the cell lines spiltted and recultured.

Response 1: Cells were passage for 2-3 times after thawing and before the start of experiments to ensure stable proliferation. For experiments cells were maintained not more than for 3 additional passages. We added this to Methods section lines 115-117.

Comment 2: What is the connection between these five cell types used in this study?

Response 2: We selected cell lines which belong to different tissues and represent some of the most common studied types. Lung cancer H1299 cells and colorectal HCT-116 cells belong to two of the most common solid cancers, glioblastoma LN-18 and neuroblastoma SH-SY5Y belong to most common cancers of neuroendocrine origin, and HEK-293 is one of the most commonly used cell lines for gene function studies, protocol optimizations, virus production, protein secretion and many others. SH-SY5Y are also widely used in neurobiology studies, as they may be differentiated into neural-like cells with adrenergic properties. The selected cell lines also grow in similar conditions, such as DMEM or RPMI-1640 with 10% FBS without the need for additional supplements, so we could provide comparison between those cell lines. We expanded reasoning for cell line selection in lines 83-92.   

Comment 3: Line 17: please provide the full name ofERK1/2.

Response 3:  We added full protein name in the text.

Comment 4: Line 55: what do the authors mean by CCCP?

Response 4:  CCCP- is a carbonyl cyanide m-chlorophenyl hydrazone, a chemical inhibitor of oxidative phosphorylation, which is used to study mitochondria dysfunctions. We added full inhibitor name to the manuscript (line 55).

Comment 5: Line 57: please add a space before the word serum.

Response 5: Thank you, we corrected it.

Comment 6: Lines 60-62: this point is really intersting. Could the authors provide more details about which genes are affected?

Response 6: These are genes involved in genes involved in glycolysis (HK2, GPI, and PFKP), metabolites transporters (SLC25A1, SLC25A11, and SLC7A1), mTOR signaling, and even translation initiation factor EIF1, and antiapoptotic factor BCL2L1. We added exact genes to the Introduction (lines 67-69).

Comment 7: Since the authors mentioned that the medium affect the replication of viruses, could the authors provide some impact about the situation with bacteria?

Response 7:  There is a limited data on how media and serum used for human cell culture could affect bacteria growth, since that would require complicated co-culture experiments. However, there is some studies which show that serum composition affects lipopolysaccharide (LPS)- and zymosan-induced procoagulant activity of mononuclear cells (Okano et al., 2006), FBS has bactericidal activity against Helicobacter pylori (Shibayama et al., 2006), probably due to proteins of complement system, and can enhance antibacterial activity of cationic and amphiphilic polymers (Sovadinova et al., 2021). We added some summary in lines 61-65.

Comment 8: Why did the authors use only a single concentration of FBS in the study?

Response 8:  Most of the cell lines are typically grown at 10% FBS and with the studied cell lines we followed ATCC recommendations. So culturing cell at lower FBS percentage may be validated for some specific experiments, but should be avoided for day-to-day cultures for reproducibility. Using higher FBS concentrations might improve cell proliferation for several brands, but again it would compromise reproducibility, and since FBS constitute the largest part of cell culture expenses it is not economically justified. As for specific conditions, which require lower FBS percentage, we actually cultured SH-SY5Y cells at 5% FBS during differentiation experiments, since the differentiation protocol requires 5% FBS. The results can be found in Figure 5c and 5d, lines 416-421.

Comment 9: All experiments were performed on cell lines, why there were no cell suspensions included?

Response 9:  One of the main aims of this study was to evaluate how growth media and FBS from different brands affect cell morphology, and most studies on cell morphology use adherent cells. Also suspension cells have different growth conditions compared to adherent cells, since they grow in volume, and adherent cells grow on surface. This makes analysis of suspension cell proliferation using automated microscopy less reliable, since suspension cells can aggregate in certain parts of the well, can float on different levels and form large clumps. We previously tried to express H2B-mRuby, which we are using for nuclei morphology, in leukemic cells, but H2B-mRuby fluorescence turns out very dim, and can hardly be detected by imaging. So inclusion of leukemic or other suspension cell lines would require using different approaches to measure cell proliferation and morphology, and those results couldn’t be directly compared with data on adherent cells. We currently work on such protocols for suspension cell lines, by to show their accuracy we need to complete a series of experiments which would not fit within one manuscript. We thank the reviewer for raising this important question and we added limitations of our approach to Discussion section lines 561-567. 

Comment 10: Was the expression of H2B-mRuby conditional or consecutive? how did the authors ensure that the cell variant phenotypes did not come because of the genetic expression of H2B-mRuby?

Response 10: H2B-mRuby expression was constitutive, as we generated cell lines using lentiviral vector, which integrated H2B-mRuby gene under human PGK promoter into cell genome. These cell lines were established previously and we showed in multiple studies that H2B-mRuby expression does not affect cell line proliferation rates or response to different inhibitors of the main signaling pathways (Mikheeva et al., 2024).

Comment 11: Line 237: please add the word "to" before the word "provide".

Response 11: Thank you, we corrected it.

Comment 12: Lines 265-267: please indicate for how long time did the cell survive without FBS.

Response 12: Typically, we maintained cells for 72h without FBS, since most experiments which require serum-free conditions do not last more than 24-48h. We clarified this in the manuscript.

Comment 13: I am still looking for a helpful conclusion of this study. What is the main message the authors want to deliver?

Response 13: Our study has several main messages, which we list in the end of discussion section. I would highlight three main messages. First, cell processes are not equally affected by the choice of media or sera. So cells, cultured using different products, might have equal proliferation rate, but that does not guarantee reproducibility in measurements of mitochondria activity or in differentiation induction. The second message: the choice of basal media, which is commonly overlooked by researches, is equally important as the choice of serum, especially for serum-free experiments. And the last practical conclusion: measuring cell morphology, even using minimalistic setup, such as bright field imaging and nuclei staining, provides valuable tool for improving reproducibility in cell culture experiments, as those measurements can capture changes in cell proliferation and various cellular processes. We added a few sentences summarizing the main message to Conclusions section, lines 574-585.

Reviewer 2 Report

Comments and Suggestions for Authors

The authors presented a fundamental study that compares cellular growth under a broad range of growth media and sera. To make the study more comprehensive, the authors included five different cell lines and assessed their growth with various tests ranging from mitochondrial activity to drug sensitivity. The amount and diversity of data collected are impressive. The introduction section is comprehensive and solid. However, the discussion and interpretation of the data are somewhat lacking. It would also be beneficial if the data presentation were more organized.

  1. On p.4 section 2.3, the authors mentioned the visualization of 123-dimensional vectors into 2D space using multi-dimensional scaling. Can the authors provide more details in this part? This method seems to contrast with more common dimension reduction algorithms such as PCA and t-SNE.
  2. On p.4 section 2.3, the authors used >100 pixels as the threshold for cytoskeleton length in differentiated cells. I would suggest that the authors justify this threshold by providing data, such as distributions of cellular cytoskeleton lengths in control and under retinoic acid.
  3. The authors claim to extract more than 100 morphological features. It would be beneficial for the manuscript if the authors could provide a list of these morphological features in the supplementary material.
  4. In Figure 2, the authors used line plots to present cellular growth rates under different media and sera. However, the lines are highly overlapping, making the error bars difficult to observe. I suggest the authors consider alternative plotting styles, such as separate bar charts.
  5. In the discussion, the authors mentioned that the results may be due to differences in media components, but they never specify what these differences are. For example, in Line 440, “…contain some additional mitogenic factors…” and Line 463, “…a result of higher cell dependency on some media components…”. It would be helpful if the authors could elaborate on what these mitogenic factors or media components are specifically.

Author Response

We thank all reviewers for thorough review of our manuscript, suggesting interesting questions, correcting some mistakes and typos, and highlighting areas which were not sufficiently covered in the initial manuscript.  Here we provide point-to-point response to all questions. All changes in the manuscript are highlighted by color.

Comment 1: On p.4 section 2.3, the authors mentioned the visualization of 123-dimensional vectors into 2D space using multi-dimensional scaling. Can the authors provide more details in this part? This method seems to contrast with more common dimension reduction algorithms such as PCA and t-SNE.

Response 1:  We agree, that current description might be confusing. Basically, we applied multidimensional scaling, which is similar in principle to PCA, but instead of commonly used metrics, such as Euclidean, we first calculated distances between data points using Wasserstein metrics. Normally when PCA or tSNE are used to compare multiple data points, each data point has n parameters, which are used for data comparison with indicated metrics, for example, Euclidean. In our case each growth condition is represented by 123 morphological parameters measured for several hundred or thousands of cells. To use any dimension reduction algorithm, we first had to summarize single cell data for each condition. The distribution of morphological parameters in cell population is not typically Gaussian, so calculating mean or median parameter values for cells under any given condition would not capture differences between cell population. Thus, we calculated Wasserstein distance, which is also called earth mover's distance (EMD). This metric depends not only on the differences between distribution medians but also depends on difference in distribution shapes. For each pair of conditions, we compared distributions for each of 123 parameters, which gave us a vector consisting of 123 Wasserstein distances. Then we calculated Euclidean distances based on those vectors, to measure dissimilarity between each pair of growth conditions. The dimension reduction was applied so we could visualize those precomputed dissimilarities in a 2D space. We updated Methods section to make this part clearer (lines 152-163).

Comment 2: On p.4 section 2.3, the authors used >100 pixels as the threshold for cytoskeleton length in differentiated cells. I would suggest that the authors justify this threshold by providing data, such as distributions of cellular cytoskeleton lengths in control and under retinoic acid.

Response 2: Thank you for this suggestion, we added Supplementary Figure S5 which shows cytoskeleton length distribution in control cells and cells differentiated with retinoic acid.

Comment 3: The authors claim to extract more than 100 morphological features. It would be beneficial for the manuscript if the authors could provide a list of these morphological features in the supplementary material.

Response 3: We added the Supplementary Table S2 which shows which parameters were extracted, which were used for dimension reduction, and which parameters were most variable in our analysis.

Comment 4: In Figure 2, the authors used line plots to present cellular growth rates under different media and sera. However, the lines are highly overlapping, making the error bars difficult to observe. I suggest the authors consider alternative plotting styles, such as separate bar charts.

Response 4: 

The reason why we used line charts was to show the cell number variance at 24h. Also, not only absolute number of cells matter, but proliferation rate (how much cells divided from 24h to 72h) also provides important information. Proliferation rates can be compared by looking at how steep the line is. We agree, that the number of data points makes it hard to review individual points and SD values. Since we also wanted to show how differences in cell proliferation translate into heatmaps we wanted to show all data points, so it would show the variance in cell proliferation. The box plots indeed allow to compare SD values for each data point, but unfortunately, they do not fit in the main figure, so we provided them in Supplementary Figures S2-S4. Also, our ShinyApp https://lebedevtdeimb.shinyapps.io/FBSMediumBrowser/ allows to select any number of datapoints, which makes it much easier to compare desired points and view SD.

Comment 5:  In the discussion, the authors mentioned that the results may be due to differences in media components, but they never specify what these differences are. For example, in Line 440, “…contain some additional mitogenic factors…” and Line 463, “…a result of higher cell dependency on some media components…”. It would be helpful if the authors could elaborate on what these mitogenic factors or media components are specifically.

Response 5: Thank you for raising this question, as we left out some details in the original manuscript. Similar question was raised by another reviewer, and we added several examples to Discussion section. Briefly, other studies indicate, that aside from differences in growth factor or hormones concentrations , FBS has variance in ATP levels, 40 elements, such as lithium, iron, magnesium and others (Bryan et al., 2011), FBS also affects secretome of cells themselves and their response to cytokines (Liu et al., 2023, Rossiter et al., 2021), and basal medium composition can affect cell dependence on glycolysis, metabolites transporters, mTOR signaling, and antiapoptotic factors (Rossiter et al., 2021). We updated the discussion section (lines 508-533, and 542-552)

Reviewer 3 Report

Comments and Suggestions for Authors

The article is devoted to the important microbiological problem of selecting nutrient media for cell growth. The authors have done quite a lot of work and presented a serious study, which, however, has a number of questions:

1. The article discusses changes in cell morphology, mitochondrial activity and drug sensitivity, but does not sufficiently explain what molecular mechanisms may underlie these differences. It would be nice to see a more detailed discussion of possible mechanisms, for example, the effect of different FBS components and media on intracellular signaling pathways.

2. The article indicates that differences in FBS composition can affect cellular characteristics, but there is no chemical or biochemical analysis of these differences. It would be great to add at least a partial comparative analysis of FBS components, for example, a comparison of the levels of major growth factors or hormones.

3. How can differences in media and FBS affect the transcriptome and metabolism of cells? Was gene expression analysis or metabolomic analysis performed to identify changes caused by different culture conditions? This could explain the mechanisms of the observed changes.

4. What is the role of mitochondrial changes in the changes in drug sensitivity? It was noted that some media/FBS altered mitochondrial activity and sensitivity to palbociclib. Is there a hypothesis related to cellular metabolic changes that could explain this effect?

Author Response

We thank all reviewers for thorough review of our manuscript, suggesting interesting questions, correcting some mistakes and typos, and highlighting areas which were not sufficiently covered in the initial manuscript.  Here we provide point-to-point response to all questions. All changes in the manuscript are highlighted by color.

Comment 1: The article discusses changes in cell morphology, mitochondrial activity and drug sensitivity, but does not sufficiently explain what molecular mechanisms may underlie these differences. It would be nice to see a more detailed discussion of possible mechanisms, for example, the effect of different FBS components and media on intracellular signaling pathways.

Response 1: 

One of the changes in intracellular signaling pathways that we identified was change in ERK1/2 activity. ERK1/2 kinases are key components of MAPK pathway, which regulates cell proliferation, survival, senescence, and even cell cytoskeleton. Although for most growth conditions changes in ERK1/2 activity were minor, still constitutive minor changes in one of the main signaling pathways might affect overall cell behavior. For example, ERK1/2 downregulation might lead to senescence, and senescent cells have increasing nuclei and cytoplasm volume. Also, medium or serum composition affects expression of EMT genes, and EMT directly changes cell morphology, drug sensitivity and many other processes. Mitochondria activity can be directly affected by variance in ATP levels, and FBS can have high variance in ATP concentrations. A recent CRISPR screen study for essential genes showed that growth conditions (medium type and serum inactivation) can determine which genes are essential for cell survival. They discovered differences in cell reliance on glucose metabolism genes, genes involved in translation, mTOR signaling, and even antiapoptotic factor BCL2L1 was essential only under certain growth condition. This shows complexity of mechanisms through which FBS or medium can affect cellular functions, and we added some of the examples to Discussion section (505-533).  

Comment 2: The article indicates that differences in FBS composition can affect cellular characteristics, but there is no chemical or biochemical analysis of these differences. It would be great to add at least a partial comparative analysis of FBS components, for example, a comparison of the levels of major growth factors or hormones.

Response 2:  We agree that in principle the knowledge of serum composition can provide valuable information, especially for development of serum substitutes. However, practically such analysis would require several types of multiplex measurements which are out of scope of our study. FBS have very complex and at the same time variable composition. Studies show that animal-derived serum can vary greatly in amount of growth factors, such as EGF, PDGF, IGF, ATP levels, microelements, such as iron. Several major biotech companies try to formulate substitutes for FBS, which include multiple components, such as. Still synthetic sera cannot fully compete with animal-derived serum, at least in quality of culturing immortalized cell lines. Several studies show that growth factors alone or in combination cannot fully substitute FBS. All these factors highlight that serum effect on cell culture comes from complex interaction of all major serum components. Unfortunately, sera producers are reluctant to provide any information on FBS composition. Moreover, it is argued that analysis of FBS composition might be not be informative since FBS composition from particular brand is usually very poorly defined, which is partially due to variations caused by animal stock breeding conditions, such as feeding, climate and season variations. Also, it many growth factors in embryonic serum have either redundant functions, or act in synergy with each other, which further complicates such analyses. Even if could identify a set of key components in serum, this information would not help to determine which FBS you should use, as FBS composition is not listed by producers. Measuring several dozens of key serum components is time-consuming and not manageable by most laboratories. So, what we propose, that high-content measurements, such as morphological profiling, provide cost- and time-effective alternative, which can capture changes in many cellular processes. We added these limitations in the Discussion section, Lines (528-533).

Comment 3: How can differences in media and FBS affect the transcriptome and metabolism of cells? Was gene expression analysis or metabolomic analysis performed to identify changes caused by different culture conditions? This could explain the mechanisms of the observed changes.

Response 3: We do not have gene transcriptome or metabolome data, as we mainly focused on what cellular processes are mostly affected by medium and serum variance, and how to access them. However, we found several studies, where authors focused on FBS/medium impact on gene expression or cell metabolism. In one of those studies authors found that different growth conditions resulted in altered levels of EMT-associated genes, namely CD44, CDH1, VIM, CD24, and in IL-8 expression along with SLC100A. Another study also showed that growth conditions determine cellular response to IL-8, and that those conditions affect cell metabolism. Moreover, FBS affects cell secretome and thus alters paracrine and autocrine loops which sustain the cell culture. Still such studies mostly provide limited insight into which exact pathways are activated by particular medium or FBS. For example, even though human plasma-like medium (HPLM) resulted in K562 cells being dependent on glycolysis, compared media had similar levels of glucose, so such dependencies may be caused by a combination of multiple factors. Since FBS has complex and highly variable composition, to identify which process are affected by which factor would require complex multivariate analysis, and thorough characterization of FBS composition, measuring all major growth factors, hormones, elements, metabolites, immunoglobulins, and etc., coupled with measuring cell response at transcriptomic, proteomic and metabolomic levels. Such study would provide priceless data for development of more reliable serum substitutes, but, unfortunately, it is out of scope of our current research goals and far beyond our financial capabilities. We updated the Discussion section (lines 505-533).

Comment 4: What is the role of mitochondrial changes in the changes in drug sensitivity? It was noted that some media/FBS altered mitochondrial activity and sensitivity to palbociclib. Is there a hypothesis related to cellular metabolic changes that could explain this effect?

Response 4: The main hypothesis here is that some drugs, such as palbociclib promote cell cycle arrest that leads to cell enlargement. Bigger cells probably require more energy and thus increased mitochondria activity, which helps those cells to survive. The proposed mechanism we described previously in (Mikheeva et al., 2024). So the hypothesis is that if media promotes mitochondria activity on it’s own, it may also facilitate survival of cells dependent on increased mitochondria activity. We speculate, that some FBS may facilitate mitochondria activation, needed for cell survival, which can be caused by difference in serum ATP levels. Serum deprivation directly affects many mitochondrial pathways, such as mitochondrial electron transport chain, oxidation of branched chain fatty acids and citric acid cycle, and FBS maintains the balance of oxidant gene expression, preventing ROS accumulation, and inhibits neomycin induced apoptosis. We included this information in the discussion section, lines (542-552).

Reviewer 4 Report

Comments and Suggestions for Authors

This is an interesting paper showing that composition of growth medium may vary depending on a product brand or lot affecting many cellular processes, and as authors presented, still those effects are poorly systematized. The authors addressed this issue by comparing the effect of 12 fetal bovine sera (FBS) 12 and 8 growth media from different brands on morphological and functional parameters 13 of five cell types: lung adenocarcinoma, neuroblastoma, glioblastoma, embryonic kidney, 14 and colorectal cancer cells. The manuscript is clearly written, and the main goal is thoroughly explained.

The methodology is well-written and includes many details that are important for its replicability and/or reproducibility. This finding certainly has important implications for the methodological field.

I have some specific comments and questions are provided below:

  1. Authors should remember about keywords.
  2. The authors conducted all analyses after 72 hours. Why were no analyses carried out after 48 hours and this time point was not presented?

Author Response

We thank all reviewers for thorough review of our manuscript, suggesting interesting questions, correcting some mistakes and typos, and highlighting areas which were not sufficiently covered in the initial manuscript.  Here we provide point-to-point response to all questions. All changes in the manuscript are highlighted by color.

Comment 1:  Authors should remember about keywords.

Response 1: We forgot to include keywords in the original manuscript text, and we thank the reviewer for noticing that. We added keywords to the manuscript.

Comment 2: The authors conducted all analyses after 72 hours. Why were no analyses carried out after 48 hours and this time point was not presented?

Response 2:  The experiment was designed so we would capture how cells adapted to changes in growth condition (24h), and how well they proliferate over time. To allow cells more time to proliferate and readapt to new components we designed the experiment so we will culture cells as much as possible. Near 72h most cell lines reach 70-80% confluence, so that allows us to analyze how well cells reached their growth end-point (at that point cell culture should be replated) and did not compromise morphology measurements (overgrown cells tend to grow onto each other). Our previous studies on cell growth dynamics show that 48h usually provides less information on cell proliferation compared to 72h (https://www.nature.com/articles/s41420-024-01950-3, https://www.jbc.org/article/S0021-9258(22)00668-8/fulltext), mostly due to the fact, that for some cell lines it takes 72h to reach optimal confluence.